# Protocol for an observational cohort study investigating biomarkers predicting seizure recurrence following a first unprovoked seizure in adults

Guleed H Adan ©,[1,2] Christophe de Bézenac,[1] Laura Bonnett,[3] Michael Pridgeon,[2] Shubhabrata Biswas,[2] Kumar Das,[2] Mark P Richardson,[4] Petroula Laiou,[4] Simon S Keller,[1,2] Tony Marson[1,2]

SSK and TM are joint senior authors.

[1]Institute of Systems, Molecular, Integrated Biology, Department of Pharmacology and Therapeutics, University of Liverpool, Liverpool, UK
[2]The Walton Centre NHS Foundation Trust, Liverpool, UK
[3]University of Liverpool Department of Biostatistics, Liverpool, UK
[4]Department of Basic and Clinical Neuroscience, King's College London Institute of Psychiatry Psychology and Neuroscience, London, UK

**Correspondence to**
Dr Guleed H Adan;
guleed.adan@liverpool.ac.uk

## ABSTRACT

**Introduction** A first unprovoked seizure is a common presentation, reliably identifying those that will have recurrent seizures is a challenge. This study will be the first to explore the combined utility of serum biomarkers, quantitative electroencephalogram (EEG) and quantitative MRI to predict seizure recurrence. This will inform patient stratification for counselling and the inclusion of high-risk patients in clinical trials of disease-modifying agents in early epilepsy.

**Methods and analysis** 100 patients with first unprovoked seizure will be recruited from a tertiary neuroscience centre and baseline assessments will include structural MRI, EEG and a blood sample. As part of a nested pilot study, a subset of 40 patients will have advanced MRI sequences performed that are usually reserved for patients with refractory chronic epilepsy. The remaining 60 patients will have standard clinical MRI sequences. Patients will be followed up every 6 months for a 24-month period to assess seizure recurrence. Connectivity and network-based analyses of EEG and MRI data will be carried out and examined in relation to seizure recurrence. Patient outcomes will also be investigated with respect to analysis of high-mobility group box-1 from blood serum samples.

**Ethics and dissemination** This study was approved by North East—Tyne & Wear South Research Ethics Committee (20/NE/0078) and funded by an Association of British Neurologists and Guarantors of Brain clinical research training fellowship. Findings will be presented at national and international meetings published in peer-reviewed journals.

**Trial registration number** NIHR Clinical Research Network's (CRN) Central Portfolio Management System (CPMS)—44976.

## INTRODUCTION
### Background and rationale

A first unprovoked seizure is a common presentation; the annual incidence is between 50 and 70 per 100 000.[1] At least 10% of the population will have at least one seizure[2] and approximately 50% will have a recurrence.[3] A major challenge is to reliably identify those

## STRENGTHS AND LIMITATIONS OF THIS STUDY

⇒ This will be the first study to prospectively investigate how brain structural and physiological architecture and connectivity in adults influences seizure recurrence following a first unprovoked seizure.
⇒ The study is expected to provide insights into the biology underlying epileptogenesis, and to lead to the development of prognostic markers of seizure recurrence following a first unprovoked seizure.
⇒ Expected recruitment has been based on records of past diagnosis and while the study is expected to recruit well, unexpected under-recruitment is possible and would be a barrier to timely completion.
⇒ A second potential limitation of this study is the potential for participant attrition and loss of patient follow-up at multiple points over 24 months; missing data could impact on the validity of study conclusions.

that will have recurring seizures, to better inform treatment decisions and counselling.[4] This will be increasingly important as we try to develop disease-modifying treatments. One paradigm will be to test new and repurposed treatments early in the disease course, using a design similar to the Multicentre Study of Early Epilepsy and Single Seizures (MESS).[5] Efficient trial designs will require the recruitment of people at high risk of seizure recurrence and this study will aid the development of EEG, MRI and blood biomarkers to do so.

Various investigations are used in the diagnostic workup of patients with first unprovoked seizures including electroencephalogram (EEG) and neuroimaging, which have been shown to have some prognostic value, but current prognostic models lack the precision to reliably stratify patients. Prognostic models of data from the MESS study[6 7] identified epileptiform EEG abnormality, neurological deficit and abnormal

MRI as significant prognostic factors. For these models, EEG and MRI were simply classified as normal, abnormal or 'non-specific'. It is now possible to use advanced quantitative approaches to analyse EEG and MRI and use them as continuous measures of neurophysiological function and anatomical variation, as proposed in this study.

In this study, we will investigate emerging epilepsy biomarkers in first-seizure populations. We already know that patients with an MRI lesion have a higher risk of recurrence,[6] however, advanced quantitative MRI analysis has never been used in this population. These methods have been demonstrated by the Liverpool Epilepsy Research Group to be effective in predicting seizure outcomes in presurgical patients with temporal lobe epilepsy.[8 9] High-mobility group box 1 (HMGB1) is a key neuroinflammatory mediator in epilepsy. Increased levels of expression of HGMB1 have been shown by the Liverpool group to be associated with increased seizure frequency in newly diagnosed epilepsy.[10] Computational analysis of resting-state EEG has been shown to reliably differentiate between cases of idiopathic generalised epilepsy and healthy controls.[11]

### Trial design
A prospective, observational cohort study that includes a nested exploratory study:
1. A prospective study in which those EEG and MRI biomarkers will be further refined and validated. Inflammatory biomarkers in the blood and saliva will also be assessed.
2. A nested exploratory study in which the utility of advanced quantitative MRI biomarkers in patients with a first unprovoked seizure will be assessed.

### Study setting
Participants will be recruited from a tertiary academic neuroscience centre in England.

### Rationale for study
#### Hypothesis
Following a first unprovoked seizure, patients at high risk of a recurrence can be identified using a combination of EEG, MRI, blood serum inflammatory biomarkers and clinical factors.

#### Overall aims
The overall aim is to undertake a multimodal investigation of brain structure, connectivity and inflammation in adults with a first unprovoked seizure. The proposed project will provide new insights into the biology underlying first unprovoked seizures in humans while also allowing us to develop prognostic markers of seizure recurrence. This research will take place in the context of collaboration between researchers with an internationally respected reputation for epilepsy research, and in an environment with demonstrated excellence in recruitment of patients with both first seizure and newly diagnosed epilepsy into research studies and clinical trials.

### Prospective substudy
#### Objectives
1. To externally validate the prognostic markers of seizure recurrence following first unprovoked seizure that has been developed in a retrospective study.
2. To identify blood and salivary biomarkers of seizure recurrence after first unprovoked seizure.
3. To identify, in a subset of 40 patients, potential biomarkers of seizure recurrence from advanced quantitative image analysis.*

*The methods described in this protocol apply to both the prospective study and the exploratory study unless otherwise stated.

## METHODS AND ANALYSIS
### Outcomes
The primary outcome event being studied is seizure recurrence following a first unprovoked seizure. We propose to use a multivariable prognostic model which evaluates the utility of multimodal biomarkers of seizure recurrence using a time-to-event outcome, with the event of interest being seizure recurrence. Seizure recurrence will be identified at follow-up intervals of 6, 12, 18 and 24 months.

### Population
One hundred patients with a first unprovoked seizure will be recruited from 'first seizure clinics' at the Walton Centre Foundation Trust (WCFT). Suitable patients will be identified by treating clinicians directly from the clinic and also from the electronic patient records. A summary of the recruitment process is shown in figure 1 and highlighted in more detail in the following section. Only patients that satisfy the inclusion and exclusion criteria below will be recruited into the study.

#### Inclusion criteria
► Aged over 16 years.*
► Diagnosis of a first unprovoked seizure (of any semiology or type, including status epilepticus) made at WCFT clinic by a member of the clinical epilepsy team.
► Maximum of eight weeks since the first unprovoked seizure.

*Recruited participants aged between 16 and 18 years will have appropriate parental consent sought in addition to their own, additional space for appropriate parental signatures will be available on the standard consent form used in the study to allow for this eventuality.

#### Exclusion criteria
► Provoked seizures (eg, alcohol or drug induced).
► Non-epileptic seizures.
► Acute symptomatic seizures (eg, acute brain haemorrhage or brain injury).
► Known progressive neurological disease (eg, brain tumour, Alzheimer's disease).

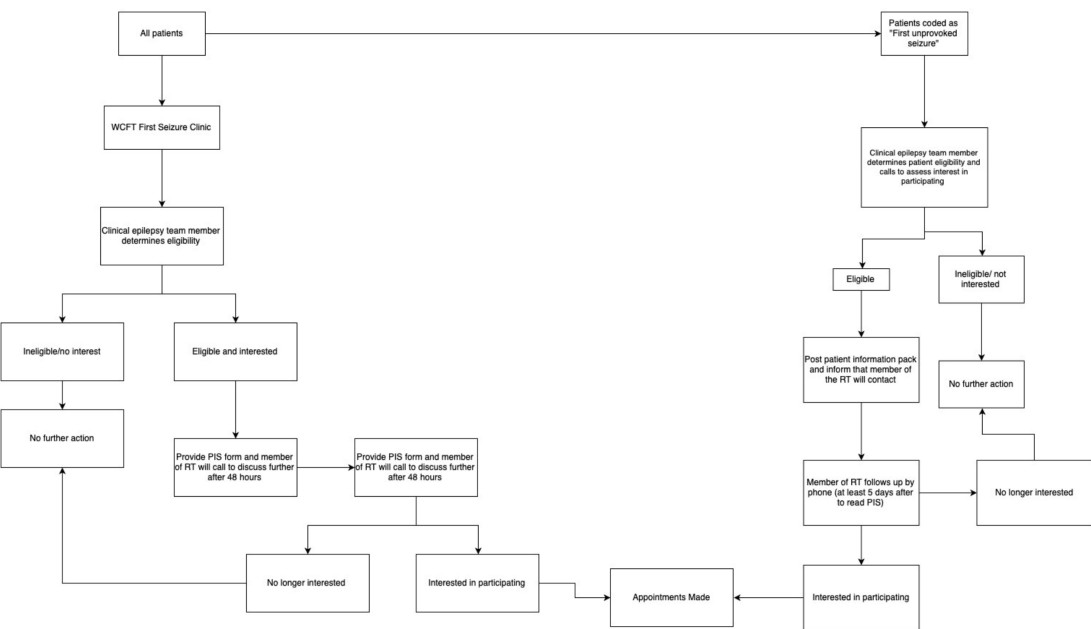

**Figure 1** - Study Recruitment Pathway. WCFT, Walton Centre Foundation Trust; RT, research team; PIS, participant information sheet.

► Known inflammatory neurological condition (specifically multiple sclerosis or sarcoidosis).
► Previous neurosurgery.
► None ambulatory patients with known significant issues with mobility which impairs the ability to independently transfer onto MRI scanner, for example, hoist transfer dependent.
► Significant medical comorbidity, for example, pre-existing severe cardiovascular or respiratory disease that would make them unsuitable for the prolonged supine positioning required for MRI scanning.
► Inability to understand written or spoken English.

### Withdrawal criteria
► Participants may withdraw their participation in this study by contacting the CI or a member of the study research team at any time.

### Recruitment via the outpatient/telemedicine clinic
A clinical member of the wider research team (ie, consultant neurologist, epilepsy nurse or neurology specialist trainee) will enquire whether eligible patients would be interested in participating in this study at the time of consultation during outpatient/telemedicine clinics.

### Recruitment via the electronic patient record system
Patients who have been coded as having a first unprovoked seizure from the WCFT electronic patient records will be assessed according to the eligibility criteria mentioned previously. Suitable patients will then be contacted by telephone by a clinical member of the epilepsy team at the Walton Centre to enquire whether eligible patients would be interested in participating in this study.

| Table 1 Summary of the procedures for each participant | | | |
|---|---|---|---|
| **Procedure** | **Location** | **Duration** | **No of examinations** |
| 1. MRI | LiMRIC, UoL or WCFT* | 1 hour, including safety examination and setup | 1 |
| 2. Blood extraction and saliva collection | LiMRIC, UoL or WCFT* | 5 min | 1 |
| 3. EEG | Neurophysiology, WCFT | 1 hour, including set up | 1 |
| 4. Telephone questionnaire | Home | 5 min | 4 (6, 12, 18 and 24 months after index event) |

*Patients in the exploratory study within the prospective cohort will have their imaging, saliva and blood sampling performed at the LiMRIC, UoL only instead of the WCFT.
EEG, electroencephalogram; LiMRIC, Liverpool Magnetic Resonance Imaging Centre ; UoL, University of Liverpool; WCFT, Walton Centre Foundation Trust.

## Procedures/assessments

Summary of the procedures for each participant (table 1):

### Study timeline

The proposed study will last 3 years and will be split into three phases, which are outlined below. The study started August 2020 with a planned end date of August 2023. We require a recruitment period long enough to recruit a sufficient number of patients with first unprovoked seizure and a follow-up period long enough to establish likely seizure recurrence.

Phase 1 (Ph1; months 1–3) is an initial 3-month period dedicated to project setup and optimisation of the MRI protocol. MRI optimisation will include technical development MRI scanning of phantoms and human volunteers to ensure the MRI sequences are adequate for the study. Necessary sponsorship and ethical approval will be sought during phase 1.

Phase 2 (Ph2; months 4–24) is a 20-month period that includes participant recruitment, baseline clinical data collection, MRI, EEG, blood and saliva acquisition for all recruited participants.

Phase 3 (Ph3; months 10–34) is a 24-month patient follow-up period during which time all seizure recurrence information will be recorded by telephone at 6, 12, 18 and 24 months after the index event.

### Sample size calculation

We propose to use a multivariable prognostic model using time-to-event outcome with the event of interest being seizure recurrence. With multiple variables of equal interest, development of a 'standard' sample size formulae is problematic. Therefore, we propose to use an event per variable (EPV) calculation. The most often cited recommendation is the rule of '10 EPV'.[12] We will recruit 100 patients, therefore, with an assumed 2-year seizure recurrence rate of 51%,[13] we would be able to include up to five predictor levels.

Forty of the 100 patients recruited will have additional advanced MRI sequences performed as part of the nested exploratory study. A sample size of 40 was chosen largely to cost limitations and given that it would satisfy the sample size flat rule of having at least 30 participants.[14] For convenience, the first 40 participants recruited will be receiving advanced MRI sequences.

### Data acquisition*

In total, we will prospectively perform clinical MRI scans, routine EEG, saliva and blood sample investigations in all of the 100 patients recruited prospectively. A table detailing data collection at the various time points of the study is presented below (table 2).

### MRI

All 100 participants recruited will have clinical MRI scans performed which will include the following sequences:
1. Conventional two-dimensional (2D) T2-weighted fast spin echo and fast fluid-attenuated inversion recovery scans, for incidental findings screening, and detection of gross pathology (together with localiser ~10 min).
2. Three-dimensional (3D) T1-weighted MPRAGE scan with isotropic voxel size of 1 mm × 1 mm × 1mm (~10 min).

As part of the exploratory nested study looking at the predictive utility of advanced imaging in first seizure patients, 40 patients will have advanced MRI scans performed (max 30 min) and will consist of:

**Table 2** Study time points

| Procedures | Baseline (T0)* | T0+6 months | T0+12 months | T0+18 months | T0+24 months |
|---|---|---|---|---|---|
| | | **Follow-up schedule** | | | |
| Signed consent form | X | | | | |
| Assessment of eligibility criteria | X | | | | |
| Contact details | X | | | | |
| Review of medical history and demographics including:<br>► Age<br>► Gender<br>► Seizure type<br>► Neurological deficit<br>► Febrile seizures<br>► Family history of epilepsy | X | | | | |
| Investigations (EEG, MRI, blood and saliva sampling) | X | | | | |
| Review of seizure occurrence by telephone | | X | X | X | X |

All EEG data collection will take place at the WCFT, in the department of neurophysiology.
*In addition to the standard clinical MRI sequences which all 100 participants will have performed, advanced quantitative MRI scanning will be performed at the LiMRIC main campus for 40 patients that are part of the cohort in the exploratory substudy outlined previously.
EEG, electroencephalogram; LiMRIC, Liverpool Magnetic Resonance Imaging Centre ; WCFT, Walton Centre Foundation Trust.

1. Conventional 2D T2-weighted fast spin echo and fast fluid-attenuated inversion recovery scans, for incidental findings screening and detection of gross pathology (together with localiser ~10 min).
2. 3D T1-weighted MPRAGE scan with isotropic voxel size of 1 mm × 1 mm × 1 mm (~10 min).
3. High-resolution diffusion kurtosis imaging (DKI) sequence with at least 60 isotropically distributed gradient directions, three b values (b=0, 1000 and 2000) and maximum voxel size of 2 mm × 2 mm × 2 mm (~10 min).

### MRI safety criteria

All participants who are having an MRI scan as part of the study will have a completed MRI safety screening as a prerequisite.

All participants will be examined by a radiographer and will complete a safety checklist that is designed to identify whether a participant has internal bodily metal, which could pose a hazard during MRI scanning. All removable bodily metal will be removed before scanning.

### Blood extraction

All 100 patients will have blood collected for analysis in a Lithium-Heparin bottles or serum separator tubes (9 mL). A maximum of 27 mL of blood (3×9 mL vials) will be obtained from each participant. Samples will be obtained by a healthcare professional trained in phlebotomy. A standard operating procedure (SOP) for blood sampling including an aseptic technique will be used by all practitioners involved in the study. Blood samples will be centrifuged within 15 min of collection or stored overnight at 4°C for centrifuge the following day. A 250 μL aliquots will then be transferred to appropriate tubes and stored at approximately −80°C prior to bioanalysis. This process is identical to other studies in the group running in parallel with REC approval (REC reference 17/NW/0342 and 19/NW/0384).

### Electroencephalogram

All 100 patients will undergo a conventional clinical EEG, using 19 channels in 10–20 arrangement, at the Department of Neurophysiology at the WCFT. Participant visiting time will last approximately 1 hour.

### Saliva

All 100 patients will have samples of unstimulated saliva collected by soaking a sponge swab in the mouth of each participant until the swab is saturated with saliva. The swab will be inserted into a collection tube. In the laboratory, the saliva sample will then be collected into an Eppendorf tube by squeezing the saturated swab using a syringe. The sample will be stored at −80°C freezer until assay.

### Data analysis
#### Clinical MRI

Analysis of clinical MRI data will be performed in all 100 participants recruited. We will perform morphometric analysis of subcortical structures, which we know

are implicated in epileptic seizures. Analysis of the data will involve using stereology in conjunction with point counting[15 16] and an automated method of volumetry for 3DT1-weighted MRI data[17] to estimate the volume of the hippocampus, amygdala, thalamus and basal ganglia in all patients.

#### Advanced MRI*
*Analysis of advanced MRI data will only be performed in the 40 participants that are included in the exploratory substudy.

(1) Thalamocortical. Preliminary data from our group has indicated that patients with newly diagnosed epilepsy who continue to experience seizures despite AED therapy have DKI alterations of thalamic projections; we will apply the same DKI approaches in our prospective study. Mean DKI values will be obtained from spatially coregistered regions of interest (ROI) (principally thalamocortical regions) in standard space. We will also apply diffusion and resting-state functional MRI independent component analysis techniques using FSL's MELODIC toolbox (http://fsl.fmrib.ox.ac.uk/fsl/fslwiki/MELODIC) and in-house Matlab scripts to identify abnormal structural and functional thalamocortical connectivity in patients relative to controls.[18–20] (2) White matter tracts. Our recent publications have indicated that analysis of white matter tract diffusion has significance for predicting postsurgical seizure outcomes in patients with chronic focal epilepsy and that DKI is more sensitive to tract pathology than diffusion tensor imaging in epilepsy.[8 9 21] As white matter tracts constitute the structural connections within brain networks, we will determine DKI properties along the length of multilobar white matter tract bundles, using our recently reported methods.[21 22] (3) Connectome. The development of whole-brain connectomes from diffusion MRI data has led to successful data-driven approaches to predict surgical responsiveness in patients with refractory focal epilepsy from members of our group.[23–26] Connectome approaches also support the association between postoperative seizure control and thalamocortical connectivity.[23] Similar methods have been applied to resting-state functional MRI data to model functional connectome alterations in chronic focal epilepsy.[27] As per our recent connectomic studies, whole-brain structural connectomes will be generated for each participant using T1-weighted and DKI data. T1-weighted data will be parcellated into multiple ROI (or nodes) using Freesurfer software (http://freesurfer.net). Structural connectivity between nodes will be determined using FSL's diffusion toolbox (http://fsl.fmrib.ox.ac.uk/fsl/fslwiki/FDT) for probabilistic fibre tracking applied to diffusion MRI. Structural connectomes will be generated using the Connectome Mapping Toolkit (http://www.connectome.ch). We will use graph theory to determine global and regional network configuration. Global network 'small worldness' will be assessed, representing the ratio between average nodal clustering coefficients and network efficiency. Regional clustering coefficient, efficiency and centrality

will also be calculated for key brain areas associated with seizure onset and propagation, such as thalamocortical and limbic networks.

### Eelectroencephalogram

The resting-state EEG activity of all 100 patients recruited will be identified by a trained clinical EEG professional. Nodes in EEG networks will be defined as electrodes, and a range of measures of interdependence between electrodes will be explored. We will apply computer models of network dynamics to resting-state EEG data.[11]

### Blood samples

Samples from all 100 patients recruited will be analysed for inflammatory markers, namely HMGB1. Inflammatory marker and HMGB1 expression analysis will be undertaken by ELISA.

### Saliva

Saliva samples will be analysed from all 100 patients to assess for concentration levels of circulating inflammatory markers including cytokine profile.

### Statistical analysis

To explore the utility of imaging, EEG and circulating inflammatory markers a series of univariable Cox regression models and a multivariable Cox regression model will be fitted for the outcome of time to next seizure following the index event, building on previous models developed with the MESS data.[6] For model development at least 10 events per candidate predictor variable is advocated.[12] With 100 patients and an assumed 2-year seizure recurrence rate of 51%,[13] we would be able to include up to five predictor levels. Backward selection, using all candidate factors (imaging, EEG and HMGB1), according to Akaike's information criterion[28] will be used to determine the parsimonious model. Bootstrap resampling with 1000 replications will be used to adjust the developed model for optimism. The accuracy of model predictions will be explored using discrimination (Harrell's c-statistic) and calibration (calibration slope) in the data used for model development (MESS data) both before and after bootstrap resampling. External validation of the optimism-adjusted model will be evaluated via Harrell's c-statistic and calibration plots. The data for the external validation are the 100 participants prospectively recruited for this study.

### Patient and public involvement

The development of the research question and outcome measures used in this study have been informed by close collaboration with local and national epilepsy charities. Patient groups were able to be consulted on their priorities, experience and preferences. Patients and their families through the Mersey Regional Epilepsy Association are involved with all research activity by our group. Patients are involved in the recruitment and conduct of the whole study. Results will be disseminated once available through epilepsy research websites, social media channels and mailing lists.

## ETHICS AND DISSEMINATION
### Ethical approval

The chief investigator has obtained approval from the North East—Tyne & Wear South Research Ethics Committee (20/NE/0078). The chief investigator will ensure a copy of the Trust R&D approval letter is available before accepting participants into the study. The study will be conducted in accordance with the recommendations for physicians involved in research on human subjects adopted by the 18th World Medical Assembly, Declaration of Helsinki 1964 and later revisions.

### Human material

Blood and saliva samples will be taken with appropriate informed consent from all 100 of the prospectively recruited participants.

Samples will be taken in the Walton Centre NHS Foundation Trust (WCFT), main outpatient building, on the day of MRI scan for the 60 participants who are having clinical MRI sequences only performed. For the 40 participants who will be having advanced brain imaging performed in addition to standard clinical MRI sequences, they will have their samples taken at the LiMRIC on the day of their MRI scan.

Blood and saliva samples will be stored in the Liverpool University Biobank (LUB) freezer room, which is housed in the Research Technology building within the LiMRIC. All samples will be stored and accessed in line with LUB SOP.

Samples of human material will be stored and archived in the LUB for 5 years following initiation of the study, after which time they will be safely and appropriately destroyed in line with standard practice. The 5-year period was chosen to allow a reasonable time frame for further analysis, following ethical clearance, after completion of the study which is expected to be within 3 years. Explicit consent will be taken from patients for the prolonged storage of this material beyond the study end date, the material will be composed of unused serum and saliva samples in case of further unspecified analysis pending ethical approval. In line with LUB SOP, the University of Liverpool will act as the custodian of the stored serum and saliva samples and explicit consent of the CI will be sought for any application to use the samples for any other purpose other than for those of the study outlined in this protocol.

### Anonymity and data governance

All EEG and MRI data will be pseudoanonymised prior to being exported from the WCFT and LiMRIC, respectively. Personal information will not be identifiable from the data. Names will be replaced with study ID numbers (PRAFUS001, PRAFUS002, etc), which can be backtracked to participant details using a key that is located

with the chief investigator, AM, who is part of the primary care team.

Digital data will be transferred from the clinical site in which it has been acquired (WCFT) to the UoL (Clinical Sciences Building, Room 2.23) for analysis on a secure, password-protected networked University of Liverpool computer.

These data will include MRI, EEG and clinical/demographic data. All data will be pseudoanonymised from the point of extraction by staff at the clinical site and transferred to GA using a secure, encrypted external hard drive—at which point patients will sequentially be allocated study IDs as per the process outlined above.

All data in the study will be in a digital format as password-protected computer files.

This will include data relating to acquired MRI or EEG information, demographic and clinical information collected both at baseline and follow-up at the various time points. All of these files will be stored on a password-protected University of Liverpool networked computer in pseudoanonymised format from the point of acquisition using the naming format highlighted above.

Pseudoanonymised digital data highlighted above will be archived on a secure, password-protected networked University of Liverpool computer located on the second floor of the Clinical Sciences Building, Room 2.23 and will be stored for a period of 20 years following study completion. Following this time, all data will be safely and appropriately deleted. Data will be kept on a secure server that offers specialised storage of many terabytes of data per project, which can comfortably accommodate all the data files generated from the study.

## Data monitoring committee
As the nature of this study was observational, it was deemed appropriate that a data monitoring committee would not be required.

## Informed consent process
All participants will be provided with a research information pack (online supplemental appendix 1) describing the nature and goals of the research, and study consent form (online supplemental appendix 2), which must be completed, signed and dated. We will not recruit participants who lack the capacity to provide informed consent (eg, those with intellectual disability or dementia). All participants will have the opportunity to discuss all aspects of the study with the investigators. Participants will have as long as they require to consider their decision to volunteer for the research or not. The investigators' contact details will be provided in the information pack. Participants are free to withdraw from the study at any time.

## Confidentiality
The chief investigator will preserve the confidentiality of participants taking part in the study and is registered under the Data Protection Act.

## Indemnity
The University of Liverpool holds Indemnity and insurance cover with Marsh UK, which apply to this study.

## Sponsor
The University of Liverpool will act as the main sponsor for this study. Delegated responsibilities will be assigned to the NHS trusts taking part in this study.

## Audits
The study may be subject to inspection and audit by the University of Liverpool under their remit as sponsor and other regulatory bodies to ensure adherence to GCP and the NHS Research Governance Framework for Health and Social Care (second edition).

## Modification of the protocol
Any modifications to the protocol which may impact the conduct of the study, potential benefit of the patient or may affect patient safety, including changes in study objectives, study design, patient population, sample sizes, study procedures or significant administrative aspects will require a formal amendment to the protocol. Such amendment will be agreed on by the research team and approved by the REC prior to implementation and notified to the health authorities in accordance with local regulations.

## Dissemination
The aim will be to publish the results in high-quality peer-reviewed journals and to present at national and international conferences. We will target epilepsy-specific events (eg, European Congress for Epileptology, International Epilepsy Congress, International League Against Epilepsy UK Chapter, American Epilepsy Society).

For each publication, only members of the research team who made a significant intellectual contribution to each piece of work will be considered as an author. This is in line with journal protocol. All authors share responsibility for the contents of the submitted manuscript.

## Final data set
The CI and members of the research team will have access to the final cleaned data set that will be stored following secure data management methods as highlighted earlier in this protocol.

**Acknowledgements** The authors are thankful to members of staff in the Institution of Translational Medicine (ITM), UoL for internal peer review of the original grant proposal and study costings for the fellowship application. The authors acknowledge all investigators at the planned recruitment site (WCFT) and Dr Kumar Das (Department of Neuroradiology, WCFT), Dr Shubhabrata Biswas (Department of Neuroradiology, WCFT), Dr Surjit Lyons-Nandra (Department of Neurophysiology, WCFT) for their support with the study. We are incredibly grateful for the feedback of the patients and their families in designing this study as well as the support of all recruited participants without whom this study would not be possible.

**Contributors** CdB, SSK and GHA contributed to the design of the MRI procedures and prepared the protocol for publication. LB contributed to the statistical elements of the study and sample size calculations. PL, MPR and MP contributed to EEG methodologies and analysis. GHA, TM and SSK contributed to the overall design of

the study, leading on the setup of data collection methods at recruitment sites. GHA, TM, SSK and CdB contributed to the analysis plan. SB, KD, CdB and SSK devised clinical and research standard imaging analysis for the study. GHA, TM and SSK conceived the study and led the development of the protocol. All authors provided critical intellectual input as per ICMJE criteria to the manuscript and have approved the final version for publication.

**Funding** This work is supported by the Association of British Neurologists (ABN) and Guarantors of Brain (GoB) via a clinical research training fellowship awarded to GA and sponsored by the UoL (6107).

**Competing interests** None declared.

**Patient and public involvement** Patients and/or the public were involved in the design, or conduct, or reporting, or dissemination plans of this research. Refer to the Methods section for further details.

**Patient consent for publication** Not applicable.

**Provenance and peer review** Not commissioned; externally peer reviewed.

**ORCID iD**
Guleed H Adan http://orcid.org/0000-0001-7340-4207

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
