## [Reviewer comments · BMJ Open]

ARTICLE DETAILS

TITLE (PROVISIONAL)	A protocol for an observational cohort study investigating biomarkers predicting seizure recurrence following a first unprovoked seizure in adults.
AUTHORS	Adan, Guleed; de Bézenac, Christophe; Bonnett, Laura; Pridgeon, Michael; Biswas, Shubhabrata; Das, Kumar; Richardson, Mark P.; Laiou, Petroula; Keller, Simon S.; Marson, Tony

VERSION 1 – REVIEW

REVIEWER	Wolking, Stefan Universitätsklinikum Aachen
REVIEW RETURNED	31-Aug-2022

GENERAL COMMENTS	Adan et al. present a protocol for an observational study to determine biomarkers for recurrent seizures after a first unprovoked seizure. They collect clinical, EEG, MRI and blood-/saliva-derived data to derive potential biomarkers. The researchers aim to include 100 persons with a history of a first, unprovoked seizure with a 24-months follow-up period. This study addresses the urgent need to better prognosticate seizure recurrence risk in this group of people and to develop parameters to identify people at risk that could potentially benefit from future anti-epileptogenic treatments. The protocol is overall concise, the methodology appears sound, and the envisioned sample size and observational period seems sensible. I have a few minor comments/questions: 1. Will all seizure types be eligible for inclusion or only GTCS?2. How is the ascertainment process that the event in question was actually a first seizure and not something else? Esp. in cases where EEG and MRI are negative? Is there in these cases a requirement for certain lab results (CK, lactate), ictal stigmata (lateral tongue bite, etc..), third-party reports..?3. Will patients with a history of febrile seizures included/excluded?4. Some persons might receive ASM treatment following the 1st seizure (e.g., if IEDs are present in EEG). Could this potentially influence the EEG and MRI parameters? How will the researchers account for this?5. What is the selection process for the 40 out of 100 that will receive advanced MRI techniques?
--

	6. How will clinically relevant findings be communicated to the participants? E.g., IEDs on study EEG that could entail a treatment recommendation. 7. The term “anonymized” is at some points used in the protocol and should be consistently replaced by “pseudo-anonymized” 8. On page 1 inclusion age is 18, later it says 16. Should be aligned.
--	--

REVIEWER	Brigo, Francesco University of verona
REVIEW RETURNED	03-Oct-2022

GENERAL COMMENTS	"A protocol for an observational cohort study investigating biomarkers predicting seizure recurrence following a first unprovoked seizure in adults." This is a protocol of a study aimed at investigating the role of combining serum biomarkers, quantitative EEG and quantitative MRI to predict seizure recurrence after a first unprovoked epileptic seizure in adults. I have the following comments:  - The methods are robust, clear and well reported. Authors could provide more details on the external validation: will it be performed in an independent cohort of patients? If so, how large will it be and how patients will be recruited? - Will patients with unprovoked status epilepticus be included as well? If not, perhaps this should be listed among exclusion criteria. - Patients with only acute symptomatic seizure(s) will be excluded, but what about patients with a first unprovoked seizure and with previous acute symptomatic seizure(s)? Will they be included or excluded? - What about patients who will be treated after a first unprovoked seizure? This will probably affect the risk of seizure recurrence. How do authors plan to address this issue?
--

VERSION 1 – AUTHOR RESPONSE

Reviewer: 1

Dr. Stefan Wolking, Universitätsklinikum Aachen

Comments to the Author:

Adan et al. present a protocol for an observational study to determine biomarkers for recurrent seizures after a first unprovoked seizure. They collect clinical, EEG, MRI and blood-/ saliva-derived data to derive potential biomarkers. The researchers aim to include 100 persons with a history of a first, unprovoked seizure with a 24-months follow-up period.

This study addresses the urgent need to better prognosticate seizure recurrence risk in this group of people and to develop parameters to identify people at risk that could potentially benefit from future anti-epileptogenic treatments.

The protocol is overall concise, the methodology appears sound, and the envisioned sample size and observational period seems sensible.

I have a few minor comments/questions:

1. Will all seizure types be eligible for inclusion or only GTCS?

All unprovoked seizures, of any type/semiology will be eligible for inclusion to minimise recruitment bias – this has been clarified in the inclusion criteria.

2. How is the ascertainment process that the event in question was actually a first seizure and not something else? Esp. in cases where EEG and MRI are negative? Is there in these cases a requirement for certain lab results (CK, lactate), ictal stigmata (lateral tongue bite, etc..), third-party reports..?

All cases recruited to the study will have had careful and detailed clinical evaluation with a clinical neurologist, sub-specialising in epileptology. Two well cited studies from our centre have found the error rates of a specialist diagnosing epileptic seizures to be between 5% and 10% (D. Smith et al 1999, Leach et al 2005). The pertinent point you raise will be carefully discussed in any resultant output from this study, however, this pragmatic study aims to replicate the reality of both the diagnostic and prognostic uncertainty faced by the clinician and patient following a first unprovoked seizure in the initial consultation.

3. Will patients with a history of febrile seizures included/excluded?

Patients with a history of febrile seizures will be included and this information will be recorded carefully in all recruited participants given the potential significance as a prognostic factor for future seizure recurrence.

4. Some persons might receive ASM treatment following the 1st seizure (e.g., if IEDs are present in EEG). Could this potentially influence the EEG and MRI parameters? How will the researchers account for this?

As the coordinating site of the MESS study (Marson et al 2005), local clinical practice is to not treat following a first unprovoked seizure unless there are features suggestive of a recognised risk of recurrence on MRI/EEG. Where this is the case, this will be logged and reported in final output from this data. In clinical practice, the greatest uncertainty is in those with seemingly 'normal' brain imaging and EEG, these patients are expected to make up the majority of recruited participants and therefore will be highly unlikely to be started on treatment following their index event. It is precisely in these cases that we will hope to develop novel biomarkers to further refine clinical prognostication and reduce uncertainty.

5. What is the selection process for the 40 out of 100 that will receive advanced MRI techniques?

For convenience, the first 40 participants recruited to the study will receive advanced MRI study.

6. How will clinically relevant findings be communicated to the participants? E.g., IEDs on study EEG that could entail a treatment recommendation.

Clinically significant results from MRI or EEG investigations will be communicated to participants in line with local and national clinical best practice, with incidental or unexpected results acted upon in a timely fashion by treating clinical clinicians.

7. The term "anonymized" is at some points used in the protocol and should be consistently replaced by "pseudo-anonymized"

This has now been corrected for consistency and accuracy

8. On page 1 inclusion age is 18, later it says 16. Should be aligned.

This has now been corrected for consistency and accuracy

Reviewer: 2

Dr. Francesco Brigo, University of verona

Comments to the Author:

"A protocol for an observational cohort study investigating biomarkers predicting seizure recurrence following a first unprovoked seizure in adults."

This is a protocol of a study aimed at investigating the role of combining serum biomarkers, quantitative EEG and quantitative MRI to predict seizure recurrence after a first unprovoked epileptic seizure in adults.

I have the following comments:

- The methods are robust, clear and well reported. Authors could provide more details on the external validation: will it be performed in an independent cohort of patients? If so, how large will it be and how patients will be recruited?

Apologies if this was unclear in the paragraph on page 10, this has been reworded to make it much clearer that model development will use the MESS data (Bonnett et al 2010) and validation will be performed using the 100 prospectively recruited participants from this PRAFUS study.

- Will patients with unprovoked status epilepticus be included as well? If not, perhaps this should be listed among exclusion criteria.

Patients presented with a first presentation of unprovoked status epilepticus will be included and this information will be recorded in all such cases given the potential significance as an independent predictor of seizure recurrence. This has now been clearly stated in the inclusion criteria for clarification.

- Patients with only acute symptomatic seizure(s) will be excluded, but what about patients with a first unprovoked seizure and with previous acute symptomatic seizure(s)? Will they be included or excluded?

Patients with previous provoked/acute symptomatic seizures that are now presenting with their first unprovoked seizure will be included, and where relevant, this information will be collected and reported.

- What about patients who will be treated after a first unprovoked seizure? This will probably affect the risk of seizure recurrence. How do authors plan to address this issue?

As the coordinating site of the MESS study (Marson et al 2005), local clinical practice is to not treat following a first unprovoked seizure unless there are features suggestive of a recognised risk of recurrence on MRI/EEG. Where this is the case, this will be logged and reported in final output from this data. In clinical practice, the greatest uncertainty is in those with seemingly 'normal' brain imaging and EEG, these patients are expected to make up the majority of recruited participants and therefore will be highly unlikely to be started on treatment following their index event. It is precisely in these cases that we will hope to develop novel biomarkers to further refine clinical prognostication and reduce uncertainty.

Reviewer: 1

Competing interests of Reviewer: None

Reviewer: 2

Competing interests of Reviewer: I have no competing interests. I have collaborated with the senior author on a prior article and on an ongoing research project.

VERSION 2 – REVIEW

REVIEWER	Wolking, Stefan Universitätsklinikum Aachen
REVIEW RETURNED	09-Nov-2022
GENERAL COMMENTS	The authors have adequately addressed all my comments/concerns.